# Comparison of Two Intergranular Corrosion Tests on EN AW-6016 Sheet Material

**Peer Decker \*, Ines Zerbin, Luisa Marzoli and Marcel Rosefort**

Trimet Aluminium SE, Aluminiumallee 1, 45356 Essen, Germany; ines.zerbin@trimet.de (I.Z.); luisa.marzoli@trimet.de (L.M.); marcel.rosefort@trimet.de (M.R.)

\* Correspondence: peer.decker@trimet.de

**Abstract:** Two different intergranular corrosion tests were performed on EN AW-6016 sheet material, an ISO 11846:1995-based test with varying solution amounts and acid concentrations, and a standard test of an automotive company (PV1113, VW-Audi). The average intergranular corrosion depth was determined via optical microscopy. The differences in the intergranular corrosion depths were then discussed with regard to the applicability and quality of the two different test methods. The influence of varying test parameters for ISO 11846:1995 was discussed as well. The determined IGC depths were found to be strongly dependent on the testing parameters, which will therefore have a pronounced influence on the determined IGC susceptibility of a material. In general, ISO 11846:1995 tests resulted in a significantly lower corrosion speed, and the corrosive attack was found to be primarily along grain boundaries.

**Keywords:** intergranular corrosion; Al-Mg-Si wrought alloys; (standardized test norms) corrosion test



## 1. Introduction

Wrought aluminum alloys from the 6xxx alloy group are extensively used aluminum materials in automotive and other engineering applications [1]. They can be used in the form of extruded profiles, forged parts such as suspension arms, or sheets for car bodies [1,2]. They have substantial mechanical properties while having good corrosion resistance. Alloys from the 6xxx group belong to the heat-treatable alloys and depend on proper thermal treatment to achieve their mechanical properties and corrosion resistance.

Intergranular corrosion (IGC) is the preferred corrosion along grain boundaries while leaving the majority of the grain intact [3–5]. It is sometimes difficult to discover but can significantly reduce the stability of a material by reducing its mechanically effective cross section. The IGC susceptibility of an alloy is dependent on its composition and thermal treatment [4,6]. For the ease of reading, the term "grain boundary" is used in the following to refer to said grain boundaries and the region close to the grain boundary that have a differing chemical composition compared to the grain center [2].

In 6xxx Al alloys, compositional inhomogeneity along grain boundaries can lead to IGC [4]. The precipitation of Si on the grain boundaries especially promotes IGC due to the difference in the electrochemical potential of Si and Al with Si acting as a cathode leading to an increased dissolution of Al [7]. The type and shape of the surface near grain boundaries also affect the extent of IGC in 6xxx alloys. If high-angle grain boundaries are dominant, IGC increases even further [8,9]. Both effects are influenced by composition, plastic deformation, and temperature. Proper thermal treatment can minimize the effect of IGC by influencing the precipitation of Si and the shape of grain boundaries [4].

The EN-AW 6016 aluminum alloy, whose main alloying elements are Si and Mg (see Table 1), is a standard alloy used in the European automotive sector for body sheet parts [10,11]. This alloy includes excessive Si above the Mg:Si ratio of 1.73:1 used for the formation of $Mg_2Si$ precipitates [12]. The additional Si enhances the mechanical properties of

an Al-Mg-Si alloy. However, excessive Si is most likely precipitated as a pure Si phase which can have a negative effect on the IGC resistance of an Al-Mg-Si alloy [13–15].

**Table 1.** Composition limits of EN AW-6016 [11].

| EN AW-6016 | Al | Fe | Si | Mn | Cr | Ti | Cu | Mg | Zn |
|---|---|---|---|---|---|---|---|---|---|
| Min (wt.%) | Bal. | --- | 1 | --- | --- | --- | --- | 0.25 | --- |
| Max (wt.%) | Bal. | 0.5 | 1.5 | 0.2 | 0.1 | 0.15 | 0.2 | 0.6 | 0.2 |

There exists a variety of different IGC tests, some developed by industrial manufacturers [16,17] and others developed by standardization organizations [18,19]. In general, IGC tests often include the exposure of the material to an acidic solution. Experience and observation of materials during a lifetime need to be considered when designing IGC tests.

Company-specific IGC tests are supposed to give a prediction on how resistant a material will be in its lifetime compared to other alloys, and they have an empirical base. This reduces the testing time from years or decades (a typical material's lifetime) to hours or days, making IGC tests applicable in modern industrial processes. The design of the test methods is crucial. If applying a not strong enough acidic solution too shortly, the material could be interpreted as too IGC-resistant. The risk of an unplanned material's fatality is then increased. On the other hand, too strong acidic solution exposure for too long a time would mean materials will fail in the test but would last in the real application. This could increase the costs by changing the application's design or utilizing a more expensive material even though not actually necessary. For this reason, it has to be repeated during production.

Norm tests, on the other hand, provide more constant results, but they may require longer testing times and more complex testing equipment. In the following, one example of each group with differing testing solutions, exposure time, and sample preparation is discussed. It is to be seen how different testing techniques will influence the evaluation of a material's IGC resistance. This work aims to illuminate this topic. It might ease the decision on what kind of IGC testing method to use in a given case.

## 2. Experiment

Sheet material made of EN AW-6016 with a thickness of 1 mm was cut into rectangular pieces with a size of 50 × 30 mm². Two different IGC tests were carried out on eight such samples. Six pieces were tested with modified versions of the ISO 11846:1995 [18]. The basic testing procedure without modifications is shown in Table 2. The other two pieces were tested following the VW-Audi PV1113 testing procedure [16]. This procedure is also shown in Table 2. The cleaning of the PV1113 samples in this work was done with acetone.

**Table 2.** ISO 11846:1995 and PV1113 testing method parameters.

| Method | Sample Preparation | Pre-Cleaning | Test Solution | Solution Amount (mL/cm²) | Exposure Time and Temperature | Reference |
|---|---|---|---|---|---|---|
| **ISO 11846:1995** | 50 × 30 mm² square with long side in rolling direction | Degreasing with acetone, 5% NaOH solution for 1 min at 60 °C, 70% HNO3 for 1 min, cleaning with acetone and DI water | 1000 mL H₂0, 30 g NaCl, 10 mL 37% HCl | Not defined | 24 ± 0.25 h at 30 ± 0.2°C | [18] |
| **PV1113** | Sample size not defined | Mandatory grinding of the edges with 120 grinding paper, degreasing with acetone or paraffin oil, ultrasonic cleaning in alcohol | 1000 mL H₂O, 20 g NaCl, 100 mL 25% HCl | Min. 8 | 2 + 0.5 h at room temperature | [16] |

The modifications to the ISO 11846:1995 were proposed by the German "Gesamtverband der Aluminiumindustrie e.V." (GDA, General Association of the Aluminum Industry) in order to better compare test results in a round robin IGC test. It includes two different solution amounts and three different acid concentrations as shown in Table 3. ISO 11846:1995 usually demands testing three samples for better statistics. In this work, only one sample per 11846:1995 variant (Table 3) was tested. The time tolerances shown in Table 2 were kept to a minimum of less than two minutes to allow better comparability of the two different test methods and the varying testing parameters of ISO 11846:1995.

**Table 3.** Varying testing parameters of the ISO 11846:1995-based tests.

| Sample Name | 1.1 | 1.2 | 1.3 | 2.1 | 2.2 | 2.3 |
|---|---|---|---|---|---|---|
| **Solution Amount (mL/cm²)** | 5 | 5 | 5 | 6 | 6 | 6 |
| **37% HCl Concentration (mL/L)** | 9 | 10 | 11 | 9 | 10 | 11 |

After cleaning, the samples were polished and optically investigated with a Zeiss Imager.A2m optical microscope (OM). The highest IGC depth was averaged over three different areas of each sample with a magnification of 100. This results in eight IGC depths (6 for ISO 11846:1995-based and 2 for PV1113).

### 3. Results

Representative micrographs for each sample are shown in Figure 1. For all ISO-11846:1995-based samples, a corrosive attack along the grain boundaries was seen. Some grains appeared to be etched out most likely because the surrounding grain boundary was completely solved. The amount of "solved-out" grains appears to be increasing with the amount of acid in the test solution. The samples tested with a solvent amount of 5 mL/cm² appeared to be showing more solved-out grains (Figure 1a,c,e) compared to their 6 mL/cm² counterparts (Figure 1b,d,f).

PV1113-tested samples showed a reduced IGC attack. However, cavities similar to a pitting corrosion attack were seen as well (Figure 1g,h).

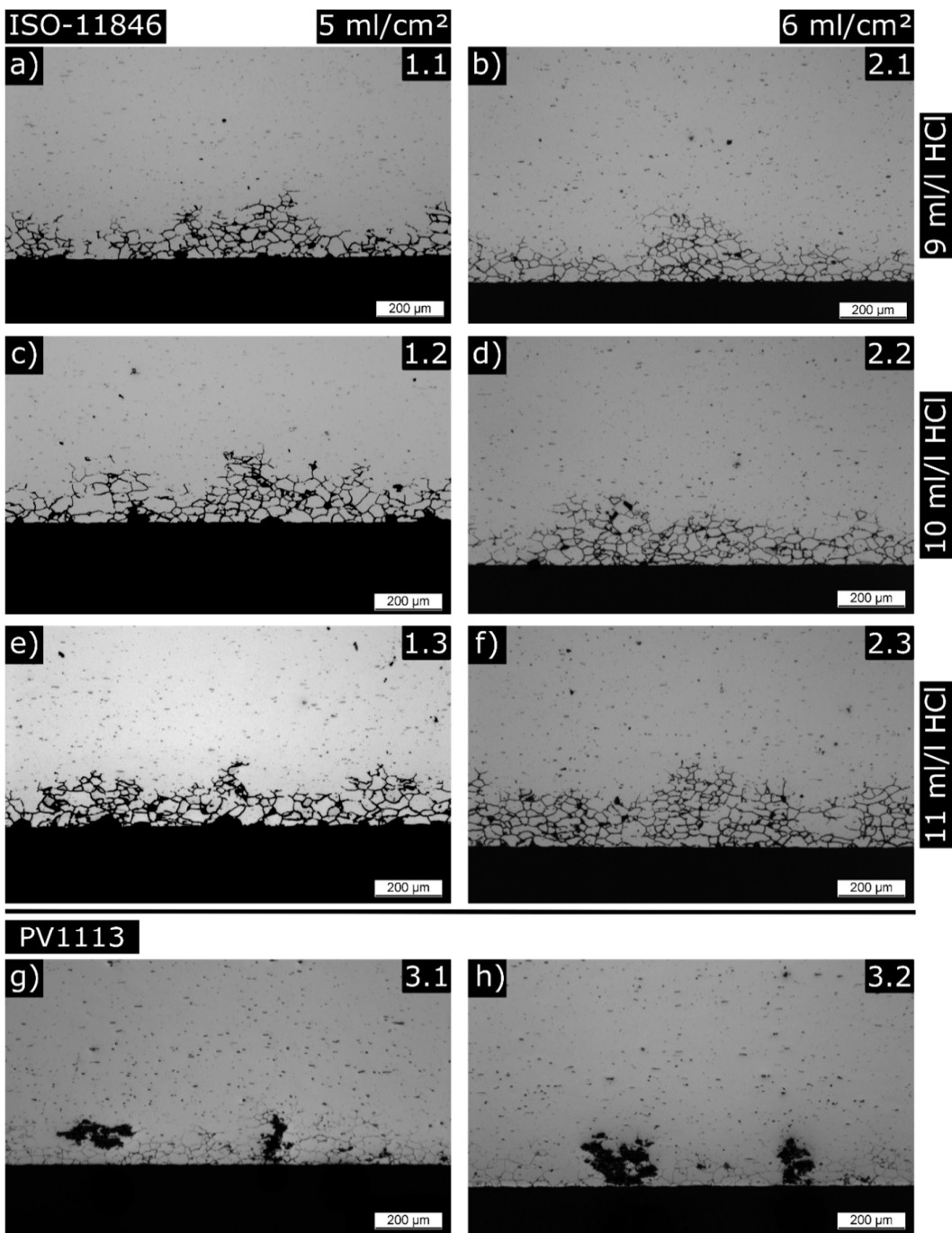

**Figure 1.** Representative micrographs of tested EN AW-6016 material. Three samples each were for the 5 mL/cm² (**a,c,e**) and 6 mL/cm² (**b,d,f**) testing solutions. Two samples were tested with the PV1113 (**g,h**).

The average corrosion depths are shown in Table 4. They vary between 173.74 μm for the 3.2 sample and 272.43 μm for the 2.3 sample. In general, PV1113 samples show a reduced IGC depth compared to ISO 11846:1995-based test samples. The minimum deviance is ~9 μm for the samples 3.1 and 1.3. The corrosion speed was determined by calculating the average corrosion depth by the respective testing time. It was found to be up to 11 times higher for the PV1113. The same holds true for the HCl/NaCl ratio. Solution amount was more than 3 to almost 3.5 higher for PV1113.

**Table 4.** Average corrosion depth and speed of the tested samples. Fractions of HCl and NaCl are also shown.

| Sample | ⌀ Corrosion Depth (μm) | ⌀ Corrosion Speed (μm/h) | Solution Amount (mL/cm²) | Test Method | vol.% HCl | vol.% NaCl | HCl/NaCl | Corrosion Type |
|---|---|---|---|---|---|---|---|---|
| **1.1** | 227.63 | 9.49 | 5 | ISO 11846:1995 | 0.33 | 1.36 | 0.24 | IGC |
| **1.2** | 249.71 | 10.40 | 5 | ISO 11846:1995 | 0.36 | 1.36 | 0.27 | IGC |
| **1.3** | 199.58 | 8.32 | 5 | ISO 11846:1995 | 0.40 | 1.36 | 0.294 | IGC |
| **2.1** | 229.44 | 9.56 | 6 | ISO 11846:1995 | 0.33 | 1.36 | 0.24 | IGC |
| **2.2** | 230.58 | 9.61 | 6 | ISO 11846:1995 | 0.36 | 1.36 | 0.27 | IGC |
| **2.3** | 272.43 | 11.35 | 6 | ISO 11846:1995 | 0.40 | 1.36 | 0.294 | IGC |
| **3.1** | 190.98 | 95.49 | 18.5 | PV1113 | 2.25 | 0.83 | 2.7 | IGC + cavities |
| **3.2** | 173.74 | 86.87 | 18.5 | PV1113 | 2.25 | 0.83 | 2.7 | IGC + cavities |

## 4. Discussion

The immediate comparison of the micrographs for both applied test methods (ISO 11846:1995-based and PV1113) partially shows a different corrosive attack. ISO 11846:1995-tested samples show primarily an attack along the grain boundaries with some surface grains being etched out. Based on the appearance of the etched-out sections, the complete dissolution of the surrounding grain boundary appears to be likelier than surface corrosion as the origin for these solved-out grains. In contrast, PV1113-tested samples show a slightly different corrosive attack. The grain boundaries are attacked as well but the formation of cavity-like features is also observed. These cavity-like features resemble more a pitting corrosion attack. It appears the more potent test solution did cause a more direct corrosive attack on the grain and not just along the grain boundary. This would explain why the IGC depth is smaller compared to ISO 11846:1995 even though the corrosion speed and HCl/NaCl ratio of PV1113 are significantly higher.

Comparing the differences in the corrosion speed, a 7–11 times higher speed is seen for PV1113 (Table 4, Figure 2). This difference suggests different potencies of the testing solutions. It is known that the solution amount (Table 4) has an impact on the extent of the corrosive attack [20], but such pronounced differences in the corrosive attack between the two tests must have another origin. A difference in the acid concentration appears to be the main reason for the pronounced differences in the corrosion speeds of both tests. The HCl concentration of PV1113 was determined to be more up to seven times higher than the HCl concentration of the ISO 11846:1995-based tests (Table 4). The NaCl concentration is the same in both tests. It is known that both HCl and NaCl concentration can influence the corrosion speed [3,21]. The determined HCl/NaCl ratio is up to 11 times higher for the PV1113 test which is almost the same factor observed for corrosion speed (Figure 2). The differences between corrosion speed and HCl/NaCl ratio result most likely from a different type of corrosive attack in PV1113 where not exclusively IGC was observed (Figure 1g,h). PV1113 tests showed the before mentioned cavities which are not recognized by the average IGC depth. It is concluded that by changing the amount of the test solution's acids or electrolyte sources, the reactiveness of an IGC test solution is changed significantly.

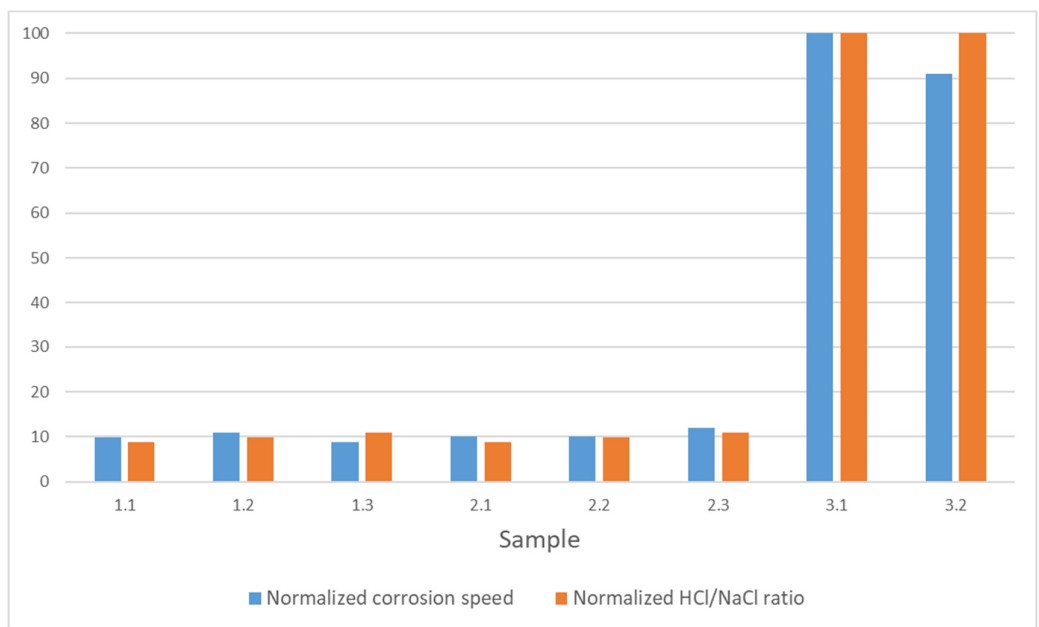

**Figure 2.** Comparison of normalized corrosion speed and normalized HCl/NaCl ratio for all tested samples.

Taking into consideration the short measurement time of PV1113 with its high temporal tolerance (2 + 0.5 h) compared to ISO 11846:1995 (24 ± 0.25 h) and the much higher corrosive speed of PV1113, a higher variance in the test results is usually expected for this test. If enough time is available, a test based on ISO 11846:1995 appears to be more reasonable to produce comparable results. If not enough time is available, PV1113 can be carried out instead, but the testing time of 2 h should be kept with lower tolerance than 0.5 h to minimize systematic test result deviation.

In this work, varying test parameters for the ISO 11846:1995-based tests were purposely made. The test solution amount was varied (5 and 6 mL/cm² each) and found to have a measurable effect on the test results (Table 5). Up to 25 % deviation for the solution amount was determined. This is in good agreement with previous observations where a higher variance of the testing method was investigated (5 and 15 mL/cm² each, [20]). The variation of the acid concentration (9, 10, and 11 mL/L) was found to have a measurable effect as well up to 20 %. These deviations are four to five times higher than the maximum observed standard deviation of the measured IGC depth of a single sample. It is, therefore, crucial to keep the solution amount and acid concentration constant to allow proper comparison, e.g., between different materials or heat treatment conditions. It is mentioned again that the solution amount is not defined in ISO 11846:1995, while for PV1113, only a minimum value of 8 mL/cm² is given.

**Table 5.** Calculated deviations of the average corrosion depths. The numbers in brackets relate to the compared samples.

| Compared Parameters | Max. Deviation (%) | Max. Deviation (μm) |
|---|---|---|
| Solution amount variation (x.1/x.2/x.3) | 1.29/7.61/26.74 | 2.95/19.13/72.85 |
| Acid concentration variation (1.x/2.x) | 20.07/16.44 | 50.13/42.99 |
| Max. observed standard deviation (1.1) | 5.9 | 13.06 |
| PV1113 variation (3.1 and 3.2) | 9.02 | 17.24 |

## 5. Conclusions

Two different IGC tests were carried out on EN AW-6016 car-body sheet material. Both tests varied in sample preparation, duration, testing solution composition, and solution amount while the determination of the IGC depth via OM was the same. Following a proposal by the GDA, the ISO 11846:1995's testing parameters were modified. The solution amount and acid concentration were varied. It was seen that a variation of these parameters resulted in deviations in the IGC depth results up to 25 %. It is therefore recommended to adhere to defined values for these parameters for a reliable comparison of test results, e.g., from different materials or different thermal treatments. It was further seen that PV1113 tests resulted in a corrosion speed up to 11 times higher than the ISO 11846:1995-based tests. These IGC speed differences were related to the HCl/NaCl ratio of the respective test solutions. The proportionalities of the IGC speeds between the analyzed samples were almost identical to the HCl/NaCl ratio proportionalities of the respective test solutions. Differences resulted from an additional pitting-corrosion-like attack in PV1113-tested samples.

PV1113 has a time tolerance of 0.5 h for the duration of the test which is twice as high as for ISO 11846:1995, even though the testing time of the latter test method is 12 times longer. Considering the much higher corrosion speed of the PV1113, a difference of ~40 µm (or ~20%) for an EN AW-6016 could appear when applying the maximum temporal tolerance. Together with a determined deviation of 9% in the corrosion depth between both tested PV1113 samples, a sum-up could result in a possible deviation of 29% in other experiments. Such a high systematical deviation makes the comparison of different tested materials difficult if not impossible. The longer duration of ISO 11846:1995 is therefore preferred to minimize the error of eventual testing time differences. The much lower corrosion speed of ISO 11846:1995 is further helpful in this regard. It is therefore concluded that ISO 11846:1995 tests should be preferred if enough time is given to carry out such prolonged testing. This would then lead to IGC results that are better comparable with each other and further allow the determination of proper IGC depth thresholds an Al alloy must withstand during testing.

It is seen that IGC susceptibility is less a material constant but is dramatically influenced by the testing conditions. This makes the comparability of different testing methods difficult. It is therefore emphasized that careful experimental planning is necessary to allow for reproducible results. Furthermore, it needs to be considered how IGC susceptibility can be evaluated when other corrosion mechanisms (as the formation of cavities in the PV1113 test) are initiated. In the given case of PV1113, it is assumed that other corrosion mechanisms occurred due to the much more potent testing solution compared to ISO 11846:1995.

This work aims to highlight concerns regarding IGC test setups and the evaluation of test results. It aims to initiate more careful thinking and planning of IGC experiments. The overall aim should be to achieve higher comparability of IGC results, which would allow for a better evaluation of a possible material's lifetime with regard to this special type of corrosion.

**Author Contributions:** Conceptulization, L.M. and P.D.; validation, L.M. and P.D.; investigation, I.Z.; writing—original draft preparation, P.D.; writing—review and editing, P.D.; supervision, M.R. and L.M. All authors have read and agreed to the published version of the manuscript.

**Funding:** This research received no external funding.

**Institutional Review Board Statement:** Not applicable.

**Informed Consent Statement:** Not applicable.

**Acknowledgments:** The authors would like to thank the GDA for providing the varying ISO 11846:1995-based testing parameters.

**Conflicts of Interest:** The authors declare no conflict of interest.

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
