# Peer review of "Comparison of Two Intergranular Corrosion Tests on EN AW-6016 Sheet Material"

_applsci, doi:10.3390/app11115294_

Round 1

Reviewer 1 Report

Abstract needs to disclose and summarise findings.

45 Cite some of these tests.

63 The objective of the research needs to be more clearly articulated. What is the problem you are seeking to solve? What would be the benefits of solving it?

68 Endnote reference error/automatic word tables?. This may be an MDPI error – one of our papers had the same problem, but just check that it is not a problem in your side.

92 “solved-out’ – try ‘dissolved’

94 ‘to be showing’ => ‘showed’

100 These mostly all look the same. Can you either quantify (imagej?) or add some arrows and text to draw attention to where you note differences.

100 Exemplary => Example

112 Please justify comments about attack of grain boundaries, given that you have not shown grain boundaries. Granted, it is difficult to optically detect grain boundaries on 6xxx due to ultrafine grains. The grains are about 20um (I quickly looked that up on https://www.researchgate.net/figure/Grain-size-distribution-of-6016-and-6014-aluminum-alloys-Fig-5-Crystal-orientation_fig4_325431407 but that might or might not be right) whereas many of your features are about 100um in size.
It is possible with special etches to see the ultrafine grains in 6xxx per https://doi.org/10.3390/met7100423 . Otherwise EBSD is good if you have it.

If you can’t get better grain definition, then you need to be more cautious in your language  where you attribute this to GBs. Obviously its plausible that the attack is along the GBs, but my point is that you have not exactly proved that.

139 The work would be improved with a speculative explanation of how the attack progresses in the two classes of corrosion. At the more fundamental level of chemistry and different composition and bond strength at GBs compared to the lattice, why do the different outcomes arise? Why cavities in the one case and cracks in the other. A bit of theoretical speculation is invited.

140 This figure does not appear to add much value beyond the data in Table 4. Either add some more comments and insights to the figure (or caption), or delete it.

162 Implications for practitioners – this is covered in 186 so no further comment.

162 Limitations of the study – briefly identify these

190 Intellectual contributions are light. Results are reported but it reads more like a routine technical report rather than a paper that moves the field forward. Try, ‘This works makes the following novel contribution to the field. It identifies that  …., and tentatively attributes this to a phenomenon whereby … causes ….’

Author Response

First of all, we would like to thank the reviewer for this thorough revision. We hope that we could address your concerns and improve the overall quality of the manuscript.

Abstract needs to disclose and summarise findings.

 A sentence was added summarizing the findings of this work.

45 Cite some of these tests.

 Added some references

63 The objective of the research needs to be more clearly articulated. What is the problem you are seeking to solve? What would be the benefits of solving it?

 Text was added to clarify the aim of the work. 

68 Endnote reference error/automatic word tables?. This may be an MDPI error – one of our papers had the same problem, but just check that it is not a problem in your side.

 It appears to be an MDPI problem, in our submitted version everything looked normal. We will contact the journal to solve this problem.

92 “solved-out’ – try ‘dissolved’

 In our opinion, “dissoveld” would mean the dissolution of the grains. In our case, we think the surrounding grain boundary was dissolved and the remaining grain was then physically removed.

94 ‘to be showing’ => ‘showed’

 Not changed.

100 These mostly all look the same. Can you either quantify (imagej?) or add some arrows and text to draw attention to where you note differences.

 The difference between the above six images is marginal due to almost similar testing methods. Differences can only be seen in the IGC depth. The two bottom images show results on the other IGC testing routine. Differences between the two groups are seen. It might be that due to the above mentioned reference error the relation between images IGC depth and text was not easy to see. We hope to have solved this reference error problem.

100 Exemplary => Example

 Not changed.

112 Please justify comments about attack of grain boundaries, given that you have not shown grain boundaries. Granted, it is difficult to optically detect grain boundaries on 6xxx due to ultrafine grains. The grains are about 20um (I quickly looked that up on https://www.researchgate.net/figure/Grain-size-distribution-of-6016-and-6014-aluminum-alloys-Fig-5-Crystal-orientation_fig4_325431407 but that might or might not be right) whereas many of your features are about 100um in size.
It is possible with special etches to see the ultrafine grains in 6xxx per https://doi.org/10.3390/met7100423 . Otherwise EBSD is good if you have it.

If you can’t get better grain definition, then you need to be more cautious in your language  where you attribute this to GBs. Obviously its plausible that the attack is along the GBs, but my point is that you have not exactly proved that.

 The grain size is depending very much on the processing techniques. The here shown grains in the range of 100 µm are not unusual for Al sheet material.

We would like to abstain from using a second etching to make all grain boundaries visible. The here presented methods are used to determine the latitude of intergranular corrosion (IGC). If a second etching would overlap the IGC test, the IGC depth could not be properly determined.

We have added a sentence in in the introduction section that is defining the term grain boundary in this work.

139 The work would be improved with a speculative explanation of how the attack progresses in the two classes of corrosion. At the more fundamental level of chemistry and different composition and bond strength at GBs compared to the lattice, why do the different outcomes arise? Why cavities in the one case and cracks in the other. A bit of theoretical speculation is invited.

 A comment on this topic was added at the end oft he conclusion.

140 This figure does not appear to add much value beyond the data in Table 4. Either add some more comments and insights to the figure (or caption), or delete it.

 We wanted to point out the significant difference between both test methods. Eventually the reference to the text is lost again due to the above mentioned field function problem

162 Implications for practitioners – this is covered in 186 so no further comment.

162 Limitations of the study – briefly identify these

190 Intellectual contributions are light. Results are reported but it reads more like a routine technical report rather than a paper that moves the field forward. Try, ‘This works makes the following novel contribution to the field. It identifies that  …., and tentatively attributes this to a phenomenon whereby … causes ….’

 We added further text to point out our intellectual intentions at the end of the conclusion

Reviewer 2 Report

This manuscript presents the research results of intergranular corrosion tests for a specific alloy material that may be meaningful for further industrial development. However, there are not many experimental results provided and the discussion is not interesting enough for publication on Applied Sciences. The recommendation for the authors is that the manuscript should be revised very carefully, added more interesting results (for example, compositional analysis before and after the corrosion, mechanical properties like hardness, toughness, strength...). In addition, all of the "reference source not found" (in the main text), please fix it carefully because this is very serious mistake when preparing a manucsript.

Author Response

We would like to thank you for reviewing our manuscript. We hope that we could answer your points in the following accordingly.

This manuscript presents the research results of intergranular corrosion tests for a specific alloy material that may be meaningful for further industrial development. However, there are not many experimental results provided and the discussion is not interesting enough for publication on Applied Sciences.

The main scope of the publication is the difference between two IGC testing methods. The material analyzed is rather common. We added text to hopefully better point out the motivation behind the work.

The recommendation for the authors is that the manuscript should be revised very carefully, added more interesting results (for example, compositional analysis before and after the corrosion, mechanical properties like hardness, toughness, strength...).

As we pointed out, the focus lies on the comparison of the two IGC testing methods. Chemical composition of the tested material was given. The chemical composition after testing is very much influenced by the chemical components of the testing solution. We believe that the chemical composition after testing will not provide any important information regarding the testing outcome. The same holds true for the mechanical properties.

In addition, all of the "reference source not found" (in the main text), please fix it carefully because this is very serious mistake when preparing a manucsript.

We found this flaw resulted from the text modification done by the MDPI. In our submitted version, these reference issues did not appear. We are very sorry for this inconvenience.

Round 2

Reviewer 1 Report

Changes noted.

The paper would be stronger if it provided a more explicit identification of intellectual contribution, or implications for future research, or implications for practitioners.  Consider adding to the Abstract some text along the lines of  'IGC susceptibility was found to be highly dependent on the testing conditions, and hence variability in protocol affects the outcomes.' As I understand it, that is the main contribution made by the paper. 

Please get the new yellow text proof-read as it has introduced new English errors.

87 Exemplary now mainly means means outstanding or excellent. Unfortunately it implies you also have bad images, but have only shown the best ones.  Perhaps use 'Typical'?

Describe the criteria used to determine corrosion depth for the two approaches, and comment on any differences.

How was average corrosion depth determined? Likewise speed.

139 This part of the work is difficult to follow. Please try again.

'In this work, varying test parameters for the ISO 11846:1995 based tests were purposely made'. How many samples?

Table 5. Which parameter is being reported here? Corrosion depth?

Author Response

We thank the reviewer again for taking the time to critically evaluate the quality of our work.

Changes noted.

Thank you.

The paper would be stronger if it provided a more explicit identification of intellectual contribution, or implications for future research, or implications for practitioners.  Consider adding to the Abstract some text along the lines of  'IGC susceptibility was found to be highly dependent on the testing conditions, and hence variability in protocol affects the outcomes.' As I understand it, that is the main contribution made by the paper. 

The Abstract was extended accordingly.

Please get the new yellow text proof-read as it has introduced new English errors.

Proof-reading was done.

87 Exemplary now mainly means means outstanding or excellent. Unfortunately it implies you also have bad images, but have only shown the best ones.  Perhaps use 'Typical'?

Exemplary was replaced by Representative.

Describe the criteria used to determine corrosion depth for the two approaches, and comment on any differences.

The criteria used to determine the IGC depth was the same for both tests and already included in the Experimental section: “The highest IGC depth was averaged over three different areas of each sample with a magnification of 100.”

How was average corrosion depth determined? Likewise speed.

Please see the answer to your previous comment. Speed was determined by dividing the average corrosion depth by the testing time. Text was modified in the results section to make that more clear.

139 This part of the work is difficult to follow. Please try again.

'In this work, varying test parameters for the ISO 11846:1995 based tests were purposely made'. How many samples?

Please see the Experimental section: “Sheet material made of EN AW-6016 with a thickness of 1 mm was cut into rectangular pieces with a size of 5030 mm². Two different IGC tests were carried out on eight of such samples. Six pieces were tested with modified versions of the ISO 11846:1995 [18]. The basic testing procedure without modifications is shown in Table 2. The other two pieces were tested following VW-Audi PV1113 testing procedure [16].”

Table 5. Which parameter is being reported here? Corrosion depth?

Correct. The caption of Table 5 was changed to make this more clear.

Reviewer 2 Report

The error of reference sources has been fixed. Some text have been added to emphasis the important of this research. The authors comment that compositional analysis and mechanical properties testing will not provide important information about the testing outcome. I must say that I do not really agree with this comment. How do we assure that the IGC corrosion type results in specimen with a better strength or hardness than the IGC + cavities type does?  
If the authors just want to focus on comparing the corrosion speed between the two tests, they should provide more relevant evidences. For example, I suggest measure the weight change/weight loss of the samples. In addition, based on the observation result (about the corrosion depth), an empirical formula about corrosion speed versus the solution amount should be provided. The scientific aspect of the research could be discussed more deeply with these findings.       

In addition, please  provide the English version of some references, for example, ref. 5, ref. 12, etc.  

Author Response

We thank the reviewer again for taking the time to review our manuscript.

The error of reference sources has been fixed. Some text have been added to emphasis the important of this research. The authors comment that compositional analysis and mechanical properties testing will not provide important information about the testing outcome. I must say that I do not really agree with this comment. How do we assure that the IGC corrosion type results in specimen with a better strength or hardness than the IGC + cavities type does?  

This work compares two different IGC testing methods. This kind of testing is a destructive one. The investigated samples are not used in any applications after testing. IGC tests try to simulate the IGC corrosion attack of complete materials lifetime. The testing solution do therefore not reflect any atmosphere a normal material would be exposed to in an application.

It is conlcueded that the mechanical properties of tested samples are not relevant to determine the IGC susceptibility of a material.

If the authors just want to focus on comparing the corrosion speed between the two tests, they should provide more relevant evidences. For example, I suggest measure the weight change/weight loss of the samples.

We appreciate your suggestion. However, as mentioned above, the scope of this work is to compare two IGC testing methods with each other. To do this, we compared the type of testing results that were defined by the norm. For the analyzed tests, it is the average IGC depth. Weight loss is a probate value to evaluate other types of corrosion and its effect on a given material. In our case, IGC susceptibility is of interest. Only small fractions of the material are dissolved. The resulting weight loss would be rather low and would be difficult to compare due to a significant potential measurement error.

In addition, based on the observation result (about the corrosion depth), an empirical formula about corrosion speed versus the solution amount should be provided. The scientific aspect of the research could be discussed more deeply with these findings.       

We suspect such an empirical formula could lead to false assumptions. The solution amount has a pronounced influence on the corrosion speed. If we would come up with an empirical formula, one could assume that the corrosion speed would only be dependent on the solution amount. Furthermore, it would only be valid for the two tests investigated in this work.

We preferred to discuss the influence of the testing parameters on the outcome of the experiments and to raise the general awareness on how the type of IGC test already influences the result.

In addition, please  provide the English version of some references, for example, ref. 5, ref. 12, etc.  

To the best of the author’s knowledge, no English version of ref. 5 and 12 exist.

Round 3

Reviewer 2 Report

This work compares two different IGC testing methods. This kind of testing is a destructive one. The investigated samples are not used in any applications after testing. IGC tests try to simulate the IGC corrosion attack of complete materials lifetime. The testing solution do therefore not reflect any atmosphere a normal material would be exposed to in an application.

It is conlcueded that the mechanical properties of tested samples are not relevant to determine the IGC susceptibility of a material.

→ Although this answer does not satisfy me, but it is reasonable.

We appreciate your suggestion. However, as mentioned above, the scope of this work is to compare two IGC testing methods with each other. To do this, we compared the type of testing results that were defined by the norm. For the analyzed tests, it is the average IGC depth. Weight loss is a probate value to evaluate other types of corrosion and its effect on a given material. In our case, IGC susceptibility is of interest. Only small fractions of the material are dissolved. The resulting weight loss would be rather low and would be difficult to compare due to a significant potential measurement error.

Agree that the weight loss can be very small, but this kind of result you can support your conclusion about the corrosion depth. In addition, though it is diffucult to compare the weight loss as you said, the measurement error can be minimized with careful prepartion and calibration of the weighing experiment. This is just my suggestion.   

We suspect such an empirical formula could lead to false assumptions. The solution amount has a pronounced influence on the corrosion speed. If we would come up with an empirical formula, one could assume that the corrosion speed would only be dependent on the solution amount. Furthermore, it would only be valid for the two tests investigated in this work.

We preferred to discuss the influence of the testing parameters on the outcome of the experiments and to raise the general awareness on how the type of IGC test already influences the result.

Again, this is just my suggestion to increase the scientific impact of your findings. An emperical formula may have two or more parameters, not only the solution amount. For example, you can come up with an emperical formula showing that the corrosion speed is influenced by the solution amount, but for the same solution amount the corrosion speeds of the two test are different. This can help to figure out a new parameter, i.e. kind of test-dependent-constant. 

To the best of the author’s knowledge, no English version of ref. 5 and 12 exist.

→ In that case, at least please provide a translation for the title, because many readers of the journal can only understand English.  

Author Response

This work compares two different IGC testing methods. This kind of testing is a destructive one. The investigated samples are not used in any applications after testing. IGC tests try to simulate the IGC corrosion attack of complete materials lifetime. The testing solution do therefore not reflect any atmosphere a normal material would be exposed to in an application.

It is conlcueded that the mechanical properties of tested samples are not relevant to determine the IGC susceptibility of a material.

→ Although this answer does not satisfy me, but it is reasonable.

We appreciate your suggestion. However, as mentioned above, the scope of this work is to compare two IGC testing methods with each other. To do this, we compared the type of testing results that were defined by the norm. For the analyzed tests, it is the average IGC depth. Weight loss is a probate value to evaluate other types of corrosion and its effect on a given material. In our case, IGC susceptibility is of interest. Only small fractions of the material are dissolved. The resulting weight loss would be rather low and would be difficult to compare due to a significant potential measurement error.

Agree that the weight loss can be very small, but this kind of result you can support your conclusion about the corrosion depth. In addition, though it is diffucult to compare the weight loss as you said, the measurement error can be minimized with careful prepartion and calibration of the weighing experiment. This is just my suggestion.   

à We agree that with careful experimentation and calibration systematical errors can be minimized. We estimated that the average loss would be in the low to medium milligram region for the given sample dimension. We estimated a rough mass loss of ~40/25/8 mg for average 250/150/50 µm attack, assuming that attack along the grain boundaries is dissolving at least 2% of the affected region. This is a rather optimistic assumption since usually not the whole depth is affected by the IGC. Furthermore, the span of the measurement values is more around 50 µm and it is the average of the deepest IGC attacks and not the overall average. The overall mass loss of the given material would then have an interval of approx. 8 mg if the measured IGC depths would be the overall average and not the deepest attack average as was done in this work. Since the deepest attack was determined the mass loss of this would be lower than the 8 mg. We suspected this low mass loss to be more susceptible to measurement errors compared to the metallographic evaluation done in this work.

We suspect such an empirical formula could lead to false assumptions. The solution amount has a pronounced influence on the corrosion speed. If we would come up with an empirical formula, one could assume that the corrosion speed would only be dependent on the solution amount. Furthermore, it would only be valid for the two tests investigated in this work.

We preferred to discuss the influence of the testing parameters on the outcome of the experiments and to raise the general awareness on how the type of IGC test already influences the result.

Again, this is just my suggestion to increase the scientific impact of your findings. An emperical formula may have two or more parameters, not only the solution amount. For example, you can come up with an emperical formula showing that the corrosion speed is influenced by the solution amount, but for the same solution amount the corrosion speeds of the two test are different. This can help to figure out a new parameter, i.e. kind of test-dependent-constant. 

  • We appreciate your suggestion and would really like to have applicable empirical formulas to predict the outcome of IGC experiments. So far, we only discussed the influence of the chemical and you are completely correct when saying that further parameters of the solvent should be included in such a formula. There is always an influence of the microstructure (e.g. density and average angle of near surface grain boundaries, chemical composition of the alloy), that would not be included in an empirical formula in this work.

We will think about an empirical formula based on your suggestion but more work needs to be done and much more (different) materials to be tested to come up with a feasible formula that would allow a reliable approximation of IGC.

To the best of the author’s knowledge, no English version of ref. 5 and 12 exist.

→ In that case, at least please provide a translation for the title, because many readers of the journal can only understand English.

  • English translations added to references